# Assessing Animal Welfare in Animal-Visitor Interactions in Zoos and Other Facilities. A Pilot Study Involving Giraffes

**DOI:** 10.3390/ani8090153

**Published:** 2018-08-30

**Authors:** Simona Normando, Ilaria Pollastri, Daniela Florio, Linda Ferrante, Elisabetta Macchi, Valentina Isaja, Barbara de Mori

**Affiliations:** 1Department of Comparative Biomedicine and Food Science, Università degli Studi di Padova, Viale dell’Università 16, Agripolis, 35020 Legnaro, Italy; linda.ferrante@phd.unipd.it (L.F.); barbara.demori@unipd.it (B.d.M.); 2Department of Veterinary Science, Università degli Studi di Torino, Largo Paolo Braccini 2, 10095 Grugliasco, Italy; ilariapollastri@gmail.com; 3Department of Veterinary Medical Sciences, *Alma Mater Studiorum* Università di Bologna, Via Tolara di Sopra 50, 40126 Bologna, Italy; daniela.florio@unibo.it (D.F.); elisabetta.macchi@unito.it (E.M.); 4Zoom Turin, Strada Piscina 36, 10040 Cumiana, Italy; valentina.isaja@zoomtorino.it

**Keywords:** giraffe, welfare, animal-visitor interaction, risk assessment, zoo

## Abstract

**Simple Summary:**

A pilot test of a six-step protocol to evaluate animal-visitor interactions was developed on a “giraffe feeding” program in a zoo. The steps devoted to animal welfare’s assessment are presented in this paper. We observed the giraffes’ behaviour, evaluated the suitability of the area in which the interactions took place, and assessed the intensity of various hazards for animal welfare. The long-term goal of this research project is to test a protocol for the overall evaluation of the quality of animal-visitor interactions in zoos and other facilities. Giraffes could choose whether to participate or not to the feeding interactions with visitors, and did not show any behaviour that suggested they did find the interaction to be a negative experience, so their welfare level was assessed as not to be adversely affected by the interaction with the visitors.

**Abstract:**

In recent years, awareness of the controversial aspects connected with wild animal-visitor interactions (AVIs) in zoos and other facilities has increased due to cultural changes. Therefore, the need to apply transparent procedures to evaluate AVIs programs in zoos and similar facilities has also increased. This study presents results of animal welfare’s assessment of a pilot test of a protocol based on six steps that aim to explore and assess the overall value of AVIs considering the impact both on animals and visitors. In the present paper, we discuss the multifaceted approach to animal welfare assessment during animal-visitor interactions, combining quantitative behavioural observations/analysis and a welfare risk-assessment procedure, which forms the basis of the six-step protocol. Pilot testing of said approach to animal welfare assessment involved giraffes (*Giraffa camelopardalis*) in an Italian zoo. No change in behaviour, suggestive of an increased welfare risk to the animals, was found. The risk analysis reported overall low risks for welfare, whereas enclosure analysis highlighted that the enclosure was suitable for allowing interactions without jeopardising animal welfare, mainly because it allowed animals to choose whether to interact or withdraw from interactions without decreasing the space available to them.

## 1. Introduction

Research, animal welfare, conservation, and education of the public are among the most important goals of modern zoos, as stated by the European Community Council Directive [1]. A possible conflict may exist between such goals, especially between the welfare of the individual animals and education/conservation [2] or between these goals and the practical necessity to make the zoo visit an entertaining experience for visitors. It has been shown that the possibility to be close to the animals and interact with them enhances the appeal of zoos and of specific animal exhibits for the public [3,4]. In turn, enhancing the emotional value of seeing animals in zoos has been shown to increases visitors’ conservation mindedness [5]. Several zoos, but also sanctuaries, rescue centres, and other facilities where animals are kept in human care, offer animal-visitor interactions (AVIs), in which visitors can have a direct (often tactile) contact with the animals. During such interactions, visitors could also learn about the species biology, ethology, and conservation. Recently, also due to a change in the way animals are perceived, awareness concerning the different, and sometimes controversial, aspects connected with such interactions has spread, leading to an increased scientific interest on the topic [6].

Several studies have shown that visitor variables (visitor presence and density being the most studied, but some studies considered also activity, size, and position) are associated with animal behavioural, and, to a lesser extent, physiological, changes [7]. Although there were situations in which the changes were suggestive of a positive, enriching experience or were ambivalent (e.g., increased vigilance, approach behaviour and vocalizations, decreased ingestion in ungulates [8]), in most cases, such changes have been interpreted in scientific literature as negatively affecting welfare [7]. For example, Anderson et al. [9] found that a higher density of petting zoo visitors correlated to increased aggressive and escape behaviours in both pigmy goats (*Capra hircus*) and sheep (*Ovis aries*). When a retreat space that visitors could not access was introduced, those behaviours were significantly reduced [9]. Baird et al. [10] found that the overall amount of handling that an animal experienced (for educational programs and for husbandry) was positively correlated both with faecal glucocorticoid metabolites and with rest, undesirable, and self-directed behaviour in zoo armadillos, hedgehogs, and hawks. Therefore, to fulfil their goals of animal welfare and conservation/education, and to increase their good reputation among the public, zoos, and facilities alike, need to apply transparent procedures to evaluate their programs of animal-visitor interactions on an integrated level. In order to do so, a careful and detailed evaluation of the effects of the interaction program both on the welfare of the animals and on visitors is an overriding priority.

In this study, we discuss results referring to animal welfare’s assessment of a six-step protocol that aims to evaluate the overall value of AVIs. The protocol has been designed to assess the value of such interactions both as educational and experiential activities for visitors and in their welfare risk valence for zoo animals. It combines behavioural and endocrine analyses with a risk assessment evaluation of both animals and visitors, an analysis of visitors’ experiences and attitudes, and an ethical standardised assessment of all the previous aspects, in order to arrive at a final step for evaluation. The final evaluation consists of a checklist to be filled in after the assessment of the interaction in order to assign a medal of a different colour that corresponds to the value of the interaction. The overall protocol is designed to help zoos and facilities to understand the strengths and weaknesses of their interaction programs and to check them periodically.

The basis of the six-step protocol consists in a multifaceted welfare assessment approach, against whose results all other possible outcomes of the interaction program, in terms of education and conservation, are evaluated. The case of a visitor-giraffe interaction in an Italian zoo was used as a pilot testing. The multifaceted and interdisciplinary approach to animal welfare assessment during AVIs proposed here combines quantitative behavioural observations/analysis and a welfare risk-assessment procedure. Results of the behavioural observations/analysis showed that giraffes did not show any behavioural changes suggestive of a negative effect of the interaction program on their welfare. The risk analysis concerning the target interaction reported overall low risks for welfare, whereas interaction analysis highlighted that the enclosure was suitable for allowing interactions without jeopardising animal welfare, mainly because it allowed animals to choose whether to interact or withdraw from interactions without decreasing the space available to them.

## 2. Animals, Materials and Methods

The pilot study was carried out between August 2017 and September 2017 in a Northern Italian zoological park (Zoom biopark, Cumiana, Turin, Italy, 44°55′59″ N–7°25′18″ E). At that time, beyond the visit to the park itself, the biopark offered visitors a range of talks, in which biologists and keepers explain the main characteristics of some of the different species housed in the park (e.g., penguins, tigers, Serengeti animals, and lemurs), and three programs involving direct contact and interactions with giraffes, tortoises, and birds of prey. The interactions with giraffes lasted 20–30 min and a maximum of 20 people could participate in each interaction after having paid an additional ticket. The study was done in accordance with both the ethical regulations of the participating institutions and the national regulations. 

### 2.1. Animals and Their Management

The study involved four male giraffes, all born in captivity, whose characteristics are detailed in Table 1.

The giraffes’ overall enclosure consists of a night enclosure and an adjacent day exhibit, divided by a ±2.5 m high fence with a gate (Figure 1). The night enclosure consists of two straw bedded (±3 m × ±1.5 m) indoor stalls and a 260 m^2^ outdoor area. The day enclosure (3600 m^2^ approximately) is a naturalistic mixed-species exhibit, housing the four giraffes, Mohr gazelles (*Nanger dama*), Thomson’s gazelle (*Eudorcas thomsonii*), waterbucks (*Kobus ellipsiprymnus*), blesboks (*Damaliscus pygargus phillipsi*), impalas (*Aepyceros melampus*), ostriches (*Struthio camelus*), and several other bird species.

During the season in which the study took place, the giraffes were lead to the day enclosure at approximately 10:30 a.m., and then the gate dividing the night and day enclosure was closed. At 06:00 p.m. (or later, depending on the evening closing time of the park), the gate was opened to allow the giraffes to be lead into the night enclosure for the night and then closed again. However, this depended on the external temperature and the climate. If it rained, for example, they remained closed in the outer part of the night area to prevent them from slipping on the wet grass and were visible to visitors. Giraffes were fed every day with oats, alfalfa, pellets for giraffes, carrots, onions, and salad in the nigh enclosure. Hay and alfalfa were always present in the day exhibit in different feeding posts, which were filled every morning. Water was provided ad libitum both during the day and in the night enclosure; in the day enclosure, there was a watercourse allowing all the animals in the enclosure to drink. The night enclosure was cleaned daily at 11:00 a.m. and the day exhibit at 08:00 a.m. when giraffes were not there.

### 2.2. The Animal-Visitor Interaction: “Giraffe Feeding”

During the season in which the present study took place, the interaction program “Giraffe feeding” was scheduled twice daily (03:20 p.m. and 04:45 p.m.) from Monday to Friday and three times a day (00:15 p.m., 03:20 p.m., and 04:45 p.m.) on Saturday and Sunday. The actual performance of each session depended on the demand of the public: i.e., if nobody booked for an interaction session, that session was cancelled. Each session lasted around 20 min. The visitors were accompanied by a keeper to the outdoor area of the night enclosure, near the fence dividing the night from the day enclosure. During interactions, the giraffes in the day enclosure were called near the fence, so that they could lean down with their neck over the fence with their head within reach of a person standing near the fence in the night enclosure. The giraffes were not forced to come when called, they could choose to come (Sam and Frantisek being the ones coming) or not, and they could interrupt and walk away from the interaction at any time. The visitors were told to stand in a line, and the keeper delivered a short talk about giraffes and gave instructions regarding how to interact with the giraffes during the program. Then, two by two visitors could approach the giraffes’ heads and feed them carrots, onions, and salad and pet them. The offered food was in addition to the giraffes’ normal diet. When all visitors had fed, petted, and eventually taken a picture of themselves with the giraffes, the interaction ended and the keeper led the visitors to exit the night enclosure. 

### 2.3. Behavioural Observations and Analysis

The basis of the six-step protocol consists in a multifaceted welfare assessment approach, which starts with an investigation of the actual welfare situation of the animals in regards to the effects of the interaction. Welfare can be investigated in different ways and using different parameters [11,12,13,14,15,16]. In this study, we chose to include the behaviour of the animals, in the form of changes in behaviour between situations in which the animals were taking part in interactions versus when they were not, as a welfare indicator. It is important to note that most changes in behaviour, unless validated, need to be interpreted as regards to their valence (expecially regarding the subjective experience of the individual) and actual meaning for the welfare status of the animal, so that they have to be considered more as risk signals, than as actual measures. This does not constitute a problem for the present study, and for the protocol, because the results of the initial welfare investigation are mainly used as a source of input for the welfare risk assessment procedure presented at Section 2.4.1, so their being risk signals is adequate to the scope they are used for. In the present study, changes in behaviour were investigated using behavioural observations and quantitative data gathering methods. These methods were chosen because they were the most represented in the scientific literature for studies on zoo animal behaviour and because they were deemed most suitable in the present study context, but other scientifically valid methods of behavioural monitoring, or even of welfare assessment, can be used in the protocol. The methods used should be adapted to the individual situation of the interaction to be evaluated. Even within the method used, the details (e.g., for quantitative data gathering, the ethogram, observation timing, sampling, and recording rules) should be also adapted to the individual situation of the interaction to be evaluated.

#### 2.3.1. Data Gathering

Behavioural observations were scheduled before (“pre-” session), during (“during” session), and after (“post-” session) the interactions and, as controls, in analogous times of the day on days in which the interaction session was not performed for lack of requests (please see Table 2 for details of the timing). With regards to the order in which the animals were filmed, all the possible permutations of sequences were listed (e.g., FKSB, FKBS, FSKB, FSBK, etc., where F stands for Frantisek, K for Karega, S for Sam, B for Baridi). Then a sequence was chosen randomly in the list for the first session, and then the following sessions followed the list of quadruplets from then on.

Seven interactions episodes (four beginning at 00:15 p.m., three at 03:20 p.m.) and seven control episodes (four beginning at 00:15 p.m., three at 03:20 p.m.), matching for environmental conditions as much as possible, were recorded. All sessions were video-recorded using a Portable HuiHeng Full HD Digital Camera (Shenzhen Weilt Electronics Co. Ltd., Shenzhen, China), following the animals in case they moved around. The videos were later observed, by a single observer, using the continuous focal animal sample method, as defined by Martin and Bateson [17], using a dedicated Behavioural Observation Research Interactive Software, BORIS [18]. For each day in which recording took place, a form was also filled recording the day, the sequence in which the animals were recorded, the start and end registration times, ambient temperature, number of participants, and other relevant events/behaviours. Temperatures were recorded in order to verify that they did not differ between control and interaction episodes, as that could be a source of bias for the results. The other data were collected so that they might be of help in case of results, which were difficult to interpret and because some of them were important for further steps in the protocol. An analysis of the effects of temperature or of the number of visitors on the behaviour of the giraffes was outside the scope of the present study: it would have needed a much bigger sample size. Some ancillary information was also gathered on the housing of the animals during the interaction, on how interactions developed, and on the animals’ history. The information on housing was derived from the management and enclosure analysis. They were concerned with whether housing allowed animals to avoid the interaction and to retreat from contact with the public without being followed, and if all animals willing to participate in the interaction could do so. Other information was gathered on how giraffes were attracted at the beginning of each interaction and on whether an initial training was needed to accustom the animals to human proximity and to take part in interactions. If training had been necessary, whether coercive methods, such as punishment and flooding (i.e., forceful exposure of the animal to possibly fear inducing stimuli at a high intensity), had been applied was investigated. The source of the animal (whether bred in captivity or born in the wild, age at which it arrived at the zoo, and, in case the animal was born free, putative age at capture) were also collected. The above-mentioned ancillary information was deemed potentially important to interpret the behavioral results. Since the welfare assessment hereby presented was designed in order to be part of a six-step dedicated protocol assessing the overall values of animal–visitor interactions in zoos or other facilities offering such interactions, the ancillary information was also deemed important in this respect.

A working ethogram was adapted from that of Seeber and colleagues [19] and finalised during the one-week preliminary observations period using ad lib observations, as defined by Martin and Bateson [17]. The only observed behaviour that may be interpreted as a stereotypy, during preliminary observations, was licking, in particular licking tree bark. Thus, no other specific entry for stereotypies was included in the ethogram (i.e., they were included in the “other behaviour” entry). However, the observer was instructed to record notes if she saw other forms of stereotypies. The same was done for other behavioural patterns not seen in the preliminary observations, but deemed of potential interest, such as other abnormal behaviour, agonistic behaviour, avoidance of contact/proximity with interacting visitors (moving parts of the body without walking away), escape attempts, and walking away from interactions. During the preliminary observation, the observer was also trained in individual recognition of the animals, and the animals could get used to the observer’s presence. The working ethogram used in the study is detailed in Table 3. The position of the animals within the enclosure was also recorded in order to verify all the space that was used. To this end, the enclosure was divided into three parts: (1) central hill (far from, but in plain view of, the visitors); (2) areas near visitors (and in plain view of them); (3) more secluded areas (far from, and not in view of, the visitors).

#### 2.3.2. Statistical Analyses

Relative durations of behaviour were calculated on the total time each animal was visible in the video recordings of each session. 

Animals, including wild animals in captivity, can markedly differ in their individual reaction to stimuli. As a consequence, a statistical approach which focuses on the individual animal (such as that used by Grisa and colleagues [20] and by Bertocchi and colleagues [21]) has been strongly advised to assess the effects of programs on the welfare of each animal involved (for example, by Aligood and colleagues [22]). Therefore, we used the same approach as in Bertocchi and colleagues [21], comparing the relative durations of behaviours performed in analogous sessions (e.g., “pre-” vs. “pre-”) in the seven interactions vs. the seven control episodes. This was done in order to investigate whether the taking place of an interactive experience with visitors had any effects on the behaviour of each studied giraffe. The percentage of time each giraffe was in each of the three parts of the enclosure was calculated and compared between control and interaction episodes in analogous sessions. Ambient temperatures were compared between control and interactions episodes. U Mann-Whitney tests were used for all the above-mentioned comparisons. A Spearman rank test was used to assess whether the number of participants and the length of interactions correlated. For all tests, alpha was set as 0.05. Inferential statistical analyses were done using the software “Statistica 64” (Dell Soft, Round Rock, Texas, TX, USA).

### 2.4. Risk Assessment

Risk assessment is a multidisciplinary, science-based process that provides an organised and logical approach for incorporating scientific information into policy development [23]. Rational treatment of the concept of risk is broad, diversified, and applicable to a multiplicity of references. Currently, the methodologies used in the fields and in the scientific disciplines that determine the risk assessments are developed at different rates and follow different paths [11,24,25,26,27,28,29]. In relation to animal welfare, the European Food and Safety Agency (EFSA) [23] developed an innovative and standardised methodology for its evaluation with a gradual approach, designed to be applied to all animal species and all factors affecting animals. EFSA has defined the risk as “a function of the probability of negative welfare consequences and the magnitude of those consequences, following exposure to a particular factor or exposure scenario, in a given population.” In recent years the number of published works on applied animal welfare in zoo animals is constantly increasing, and particularly on the welfare consequence of factors like human impacts [10,30,31,32]. The environmental conditions of zoos where educational and entertainment activity with an animal-visitor interaction occur, with the consequent exposure to different hazards during accomplishment of single activities, justify the need of a specific evaluation of the risk. For this purpose, we provided the analysis of the risk related to animal-visitor interactions in order to understand and manage the possible consequences that could occur with this activity. The application of risk assessment to zoo animal welfare issues is a relatively recent approach, and, to our knowledge, risk assessment has never been applied before to welfare in relation to animal–visitor interactions, so its implementation in this context, within a multifaceted welfare approach, deserves to be examined in detail.

#### Risk Assessment Related to Animal Welfare

The assessment was carried out following the methodology provided by the EFSA guidelines [23]. The evaluations of the risk were done using semiquantitative methods: the evaluations were carried out in qualitative terms and subsequently transformed into numbers to be processed in calculation algorithms. Risk cannot be characterised by numbers alone. For many scientists, this means a less precise analysis and lack of scientific rigour, as quantification is superior to qualitative analysis in terms of suitability for generating knowledge that enhances our understanding of cause-effect relationships [33]. For some scientists, it also means more subjectivity, as they consider quantification less subjective than qualitative analysis. However, choosing a purely quantitative approach brings challenges in relation to properly representing and treating all types of risk and uncertainties [34]. The problem is, thus, either sticking to a quantitative approach, which has strong limitations, or using a combined semiquantitative approach, which seeks to meet these limitations. The authors see no alternative to the latter approach, though they acknowledge that a certain degree of uncertainty, and thus subjectivity, is unavoidable when dealing with probability, predictions, and risk evaluation in general. 

The first step was to define the problem formulation by clarifying the risk question, identifying the target population, the factors of animal welfare concern, the exposure scenarios, the known animal welfare consequences and their measurement, and by building a conceptual model.

Risk Question: “What are the consequences on welfare if a management model that provides for a direct human-animal contact is introduced in comparison to a management model where that contact is absent?”

Target population: four individuals of the species *Giraffa camelopardalis* (see Table 1).

Identify factors of animal welfare concern: the interaction itself was identified as the risk factor that could lead to animal welfare change. This and other management factors have the potential of causing unpleasant subjective experience, injuries, or diseases. As a matter of fact, the design of the enclosure where the interactions are offered and the management aspects can strongly affect both animal welfare and visitors’ experience. Therefore, a management and enclosure analysis centered on aspects connected with the interaction activities must be considered in an interdisciplinary assessment of animal-visitor interaction. A checklist was created (Table 4) to investigate how adequately the enclosure could maintain a high standard of welfare during interactive activities and if management was suitable to prevent risks for the welfare of the animals. The checklist was designed to be filled by the researcher, but can be used also by the staff that are in charge of assessing the quality of the interaction. The results are integrated into the risk assessment analysis.

Identify exposure scenarios: an exposure scenario is a sequence or combination of events in relation to the risk question that includes, in general, all information on the events to which animals of the target population are subjected. In this phase, relevant combinations of the identified factors and their exposure levels are defined. The exposure scenario assumed that the new management model could negatively affect animal welfare with consequences identified as behavioural alterations following the worsening of subjective experiences (scenario 1), traumatic injuries (scenario 2) and infections (scenario 3). 

Identify the known animal welfare consequences and their measurement: welfare consequences are changes in welfare that result from the worsening of the subjective experiences of the animals. The welfare consequences considered in this risk assessment study were: behavioural alterations, injuries, and infectious disease. Animal-based measures (indicators) are necessary to assess the welfare consequences, as they measure qualitatively or quantitatively the worsening condition of welfare and their interpretation and assessment will depend on their magnitude.

Build a conceptual model: a conceptual model in problem formulation is a description of a model of known or supposed relationships between factors (welfare hazards) and animal welfare consequences (expressed by animal-based indicators). For the construction of the conceptual model, it has been hypothesised that the probabilities of occurrence of all the considered consequences on welfare are negligible in the management model without interaction. The conceptual model that we have developed includes three different scenarios. The first two scenarios recognize a common factor. We have defined that a wrong handling or an improper approach are factors that can determine behavioral alterations that could indicate the perception of negative subjective animal experiences (scenario 1) and that could also provoke injuries (scenario 2). Scenario 3 defines as exposure factor an effective contact with a zoonotic agent that can cause an infectious disease. 

Once the problem formulation was defined, the next stage was proper animal welfare risk assessment that comprises three steps: (1) exposure assessment; (2) consequence characterization; and (3) risk characterization.

The exposure assessment provides an evaluation of the strength, duration, frequency, and patterns of exposure or the factors relevant to the exposure scenarios developed during the problem formulation.

In the exposure assessment phase, we tried to estimate the frequency of exposure (FE) to the factors defined in the various scenarios using a semiquantitative method where the categories negligible, low, moderate, and high take numerical values on an ordinal scale from 1 to 4. Negligible describes an event very unlikely to happen, not expected to happen, and with <5% chance of happening, low is an event not expected to happen, but it may occur with a probability ranging from 5% to 30%, moderate is expected to happen with a probability ranging from 31% to 70%, high is an event that will occur with a probability >70%.

The next phase included consequences characterization and defining the categories of observations and records to quantify the magnitude (expressed by the product of severity of the consequences multiplied by their duration) and the probability of their occurrence. The evaluation of the probabilities is given by the frequency with which the consequences (FC) analysed are manifested over a pre-established period of time (e.g., duration of the study). FC is expressed in a semiquantitative method using the same categories and the same probability range as for the FE.

Animal-based measures (indicators) are necessary to assess the welfare consequences, as they measure the worsening condition of welfare (severity). For the duration, categories are defined according to a semiquantitative method that assigns values proportional to the progression of the event (in which a time frame of days assumes the value of 1 and a time frame of months assumes value 2).

Three indicators were created to quantify the severity of the consequences identified in the different scenarios (Table 5, Table 6 and Table 7). For the construction of the indicators we were inspired by the Welfare Quality^®^ project and the scheme of presentation is faithful to the Welfare Quality^®^ Assessment Protocols [35].

The output was measured as welfare score (WS) values obtained from the following formula:

WS (welfare score) = FE (Frequency of exposure to the factor in a specific scenario) × MA (Magnitude as a product of the severity of the consequences for their duration in a specific scenario) × FC (Frequency of consequences in a specific scenario)

As the risk assessment needs to meet the needs of the decision making, to support it we elaborated a table that has the function of correlating the values of WS we obtained with specific measures of risk mitigation. In fact, starting from a stronger knowledge about the relevant factors influencing exposure and consequences, given a risk ranking, the jump to measures which can reduce the risk is quick (the knowledge aspect of risk is the key to reduce it) [36].

So the WS values obtained were compared with those present in a specifically designed table for the determination of the corrective actions to be undertaken (Table 8). The values of the welfare score and the related actions to be taken derive from combinatorial simulations between the exposure elements and the consequences in order to provide management measures compatible and consistent with built scenarios, in accordance with the “as low as reasonably practicable”—ALARP (As Low As Reasonably Practicable) principle [37].

## 3. Results

### 3.1. Behavioural Observations and Analysis

Being fed by visitors during the “during” sessions significantly differed in relative duration between interaction and control episodes for the two giraffes taking part in interactions (Z = −3.07; *p* = 0.002 for Sam, Z = −2.17, *p* = 0.03 for Frank, Appendix A available online at http://www.mdpi.com/2076-2615/8/9/153/s1). As is logical, the giraffes could only be fed by visitors during the interaction, whereas they could not during the control episodes. A “total ingestion” relative duration was also calculated adding up, on the excel file, the relative durations of Browse + Feeding + Drinking + Being fed by visitors + other ingestion behaviour, in order to investigate whether getting food from visitors substituted or added to the other ingestion behaviours. Such “total ingestion” also differed in relative duration in the “during” sessions between interaction and control episodes for the two giraffes taking part in interactions (Z = −2.81; *p* = 0.005 for Sam; Z = −2.30; *p* = 0.02 for Frank). The giraffes showed total ingestion behaviour longer during interaction vs. control episodes. No other behaviours showed any statistically significant difference during the “during” sessions. No behaviour differed in the “pre-” and/or in the “post-” sessions between interaction and control episodes. No differences in behaviour were detected for the two giraffes not taking part in the interaction programs between the interaction and the control episodes. Apart from licking, no other behaviour that could be a form of stereotypy was recorded. No abnormal behaviour, agonistic behaviour, escape attempts, avoidance of contact/proximity with interacting visitors, or getting away from interactions were recorded during observations either. 

All the giraffes were recorded spending time in all the areas of the enclosure. There were no significant differences in the percentage of time spent in any of the three parts of the enclosure by any of the giraffes in any of the sessions between interaction and control episodes. Ambient temperature was not different between the interaction and control episodes (Z = −1.5; *p* = 0.13). People participating in interactions varied between 9 and 23 (median 18), and interaction duration varied between 14 and 21 min (median 15.5 min), with no correlation between number of participant and interaction duration (R = −0.41; *p* = 0.36). Information gathered on how the two giraffes taking part in the interactions were housed revealed that they were free to interact or withdraw from them. No coercion was used to attract them at the beginning of each interaction, and there was no reason to infer that any form of coercion was used to habituate animals to humans and during initial training. 

### 3.2. Risk Assessment Related to Animal Welfare

#### 3.2.1. Identify Factors of Animal Welfare Concern

The results of the checklist for management and enclosure analysis showed that management and enclosure were fairly adequate to maintain a high standard of welfare during interactions. The only items to which a negative answer was recorded were:Those concerning washing and disinfection of the visitors’ hands before the interaction in order to minimize the possibility of pathogens transmission from people to the animals (items 4, 29, 30), andThat about reminding the visitors of the rules of the interaction (item 27).

#### 3.2.2. Exposure Assessment

For scenario 1 and scenario 2, which have different consequences (behavioural alterations indicating that animals were at risk of perceiving negative subjective experiences and injuries), but recognize the same exposure factors (wrong handling/improper approach), it is possible to quantify the frequency of incorrect human behaviour patterns using the analysis of the videotapes made during the interaction session (“during” observation session). Since no different behaviors were observed, the probability is considered negligible. Concerning scenario 3, we defined this scenario as having a low probability of effective contacts. Even when visitors wash and disinfect their hands before the interaction activity (see the Checklist for the management and enclosure analysis Table 4), spreading of some pathogens by aerosol may be possible in absence of preventive measures. Even if no evidence of pathogen spreading consequences were observed during the study, the probability was still considered low, as a precautionary measure.

#### 3.2.3. Consequence Characterization

In this phase, we evaluated the severity of the consequences resulted from the exposure to the three different scenarios (Table 5, Table 6 and Table 7). In our study, no consequences on welfare were observed, but as a precautionary measure, the probability has been defined as low and not as negligible. In fact, we cannot exclude, due to a short period of observation, the possibility that there could be negative consequences for welfare in the future.

#### 3.2.4. Risk Characterization

The risk characterization is then obtained by integrating the exposure assessment with the characterization of the consequences usually expressed multiplicatively with a welfare score (WS) ranking: WS=FE×MA×FC

Thus, welfare scores were calculated (Table 9):

Welfare score associated with scenario 1: FE1 frequency of exposure × MA1 magnitude of consequences for scenario 1 × FC1 probability that animals experience unpleasant subjective experiences (value derived from behavioural data analysis).

Welfare score associated with Scenario 2: FE2 frequency of exposure in scenario × MA2 magnitude of consequences × FC2 probability that animals were subjected to injuries (value derived from the external visual examination).

Welfare Score associated with Scenario 3: FE3 frequency of exposure × MA3 magnitude of consequences × FC3 probability that animals were infected (value derived from diagnosis of infectious disease).

The final values of welfare score were then compared with a reference table (Table 8—welfare risk assessment score) for the evaluation and comparison of the different management mode.

## 4. Discussion

In this study, we focused on the implementation of a multifaceted welfare assessment approach, which constitutes the basis of a six-step protocol for AVIs’ evaluation. The aim of the overall protocol is to evaluate the quality of AVIs delivered in zoos and other facilities, whatever the species involved. The various steps of the protocol can be partially adapted from time to time to the different species, the degree of interaction, and the management each facility is willing to adopt. Data presented in this paper are the ones collected in a first pilot test of the protocol regarding an interaction where there was a direct contact with visitors and where the species involved was a mammal.

In the present giraffe’s pilot study, the implementation of the behavioural observation and analysis had to take into account both organizational and scientific problems, which are likely to be common to other AVIs assessments. Although all the staff at the zoo were very supportive, it was difficult to know with suitable advance whether and exactly at what time an interaction would take place. Also, in periods in which the public demand for interaction was high, it was difficult to find a control episode to video-record. From a scientific point of view, another limitation to this type of assessment procedure is the virtual absence of scientifically validated behavioural parameters representing decreased welfare for each of the many species that can be involved in AVIs. This is true for the present pilot study on giraffes, even if this species is often involved in AVIs. To our knowledge, no study has validated specific stress-related behaviours in giraffes, and few studies have focused on behaviours suggestive of stress in giraffes. For example, Tarou et al. [38] found increased levels of activity, stereotypical behaviour, and contact behaviour (particularly neck-rubbing) in two female giraffes when a male, that was usually housed with them, was removed. As social separation is a known stressor in social species, the behavioural changes identified by Tarou et al. [38] during separation are likely to be linked to negative subjective mental states in the studied animals. Siciliano-Martina and Martina [39] also found that giraffes showed more tongue playing when the ambient temperature was low, attributing the finding to increased thermal discomfort. Mason [40] suggested that time spent entirely inactive could be indicative of stress and to sometimes replace stereotypes, but Siciliano-Marina and Martina [39] did not find a difference in activity between hand-raised and maternally raised giraffes, leading them to conclude that one should not generalize inactive periods as stereotypic or stress related among giraffes. In the present pilot study, no changes in the above-mentioned behaviours were detected, so it is reasonable to say that behavioural observations/analysis did not identify changes in the giraffes’ behaviour suggestive of altered risk to the welfare level of the animals. Moreover, the way in which the interaction took place also lends strength to the hypothesis that it is unlikely that the giraffes were stressed. It has long been known that, if animals have the possibility to control events, they tend to not perceive them, even negative ones, as stressors, as noted by Mason and Mendl [41] and Dess and colleagues [42]. In the case of the interaction program assessed in the present study, the giraffes taking part in the interaction chose to do so and were not forced to interact in any way, hinting that they perceived the situation as positive. The interaction consisted mainly of giraffes being offered very palatable (but deemed relatively low calorie) food, which was in addition to their normal diet, so the giraffes were very unlikely to have a reason to participate if they perceived the situation as negative. In the present study, there were no differences in licking (the only behaviour expressed by the giraffes involved in the present study, which could have been interpreted as a stereotypy) and resting standing (i.e., being idle) between interaction and control episodes. The above-mentioned finding differs from what found by Orban et al. [43]. Several factors could have contributed to the difference, highlighting the importance of customising assessment to the specific AVI to be evaluated and to interpret results accordingly. The studies had different scheduling of observations, which were spread throughout the day in Orban et al. [43] vs. concentrated during and immediately before and after the interaction in the present study. Therefore, behaviours and changes which took place outside the times chosen for recording could have gone undetected in the present study. Moreover, Orban et al. [43] compared the behaviour of giraffes who were in zoos differing in the amount of interaction offered (i.e., full day, part day, and no interaction), whereas, the present study, the behaviour of the same giraffes was compared between interaction and control episodes. 

It would be interesting, as the goal of a future study, to investigate why two giraffes never chose to participate to interactions, as it could be due to several reasons, some of which might have relevance to welfare. Although the area used for interaction allowed for all four giraffes interacting with visitors without being in physical contact among them, it might not have been big enough for submissive animals to feel that they were a safe distance from other giraffes, especially given the presence of resources. If this was the case, it might have been the source of frustration for the giraffes unable to interact. In many zoos, it is not practically feasible to organise contemporary feeding experiences for all animals of the same enclosure so that they are as distant as to exclude that the presence of one giraffe could inhibit another from interacting. However, the absence of differences in behaviour and in the use of the different parts of the enclosure between control and interaction episodes for the two giraffes not participating appears to disagree with such hypothesis, as does the virtual absence of agonistic behaviour among all the giraffes. Another explanation could be that individual giraffes differed in the subjective valence they attributed to what happened in the interactions, so for some of them obtaining extra food was not worth participating in interactions with people.

As far as the methods chosen for behavioural data collection were concerned, they were found to be satisfactory for the goal for which they were intended. Their main use of behavioural data in the protocol was to be fed to the risk assessment procedure as the animal-based indicators of possible welfare risks. As the relevant behaviours to this end could be short time events, the choice to use a continuous sampling method over an interval one made it unlikely that such behaviours went undetected. It is important to note that, even within a quantitative behavioural evaluation, the actual behavioural parameter that can be useful to be entered in the risk assessment are somewhat flexible depending on the situation. Some, such as escape attempts, agonistic behaviour, and abnormal behaviour, are likely to be used in most assessments, but others can vary per se or in the way they are gathered. For example, in the present pilot study, where visitors were not in the same enclosure as the animals, space use was collected dividing space in broad areas, differing in visibility and nearness to the public. In another study, where Aldabra giant tortoises (*Geochelone gigantea*) were involved in interactions, during which the visitors were allowed inside the enclosure where the tortoises were, a parameter of space use which was fed to the risk assessment procedure was the use of specific relatively small cave shelters, where people had difficulties in reaching the animals (data not shown). In the present study, no data on latency was deemed important, however, in a situation in which some form of environmental enrichment is distributed during interaction time, latency to approach the enrichment items could be important as an animal-based indicator of possible welfare risks. In AVIs in which only some members of a group participate, and are separated from the rest of the group in order to do so, a detailed study of vocalizations, also using phonograms, could be needed.

This paper also focused on the application of the risk assessment methodology to compare three different scenarios affecting the welfare of zoo animals during the interaction program. The application of risk assessment procedures is a relatively novel approach to animal welfare in zoos, even more so when applied to the effects of AVIs. The advantage of choosing a semiquantitative method is due to the fact that it can be used to aid decision-making in circumstances in which there is a lack of specific knowledge about how uncertainty can affect the assessment. A longer period of observation would have been useful to decrease the level of uncertainty both about the probability of pathogen spreading in the exposure assessment and about the probability that giraffes will develop diseases or other negative consequences on welfare. In the present pilot, such uncertainties were reduced using the precautionary principle (i.e., choosing a level slightly higher than that resulting from the study) and making the assessment flexible to deal with the possibility of underestimating the risk. 

In the problem formulation phase, risk management options were considered using a checklist for the management and enclosure analysis in the process of assessing the risk (Table 4). In the same way, a specifically designed table for determination of the corrective actions that could be undertaken (Table 8) was introduced in the risk characterization phase. In this way, the conceptual separation between risk assessment and management was lost. However, this was deemed not to introduce undue bias into the process and allowed for the avoidance of detrimental effects on the utility of the generated output (in that risk assessors and risk managers often find difficult to work together); adopting this decisionistic model rather than the technocratic one based on the independence between assessment and management. This phase makes a judgment call on whether or not a risk is acceptable or whether specific risk reduction measures are necessary.

In all three scenarios, the result in welfare score determines an acceptable response outcome. Therefore, with respect to the management model in the absence of interactions, there are no differences in terms of possible necessary corrective actions. The welfare risk analysis identified an overall low risk for the welfare of the giraffes: in scenario 1 this was mainly due to the absence of significant behavioural changes and the fact that the giraffes could choose whether to interact or not and in scenario 2 this was due to the absence of lesions. In scenario 3, this was due to the risk associated with zooanthroponosis resulting in a welfare score of “four”, and therefore being at a threshold level of attention, a depth epidemiological investigation is recommended in the case of passive surveillance (permanent veterinary control) leading to the finding of any zoonotic disease.

Moreover, the application of a risk assessment methodology to an animal welfare assessment in a zoo highlighted its potential use for reviewing the management decision plan.

To better safeguard the health of the animals taking part in interaction by reducing the possibility of transmission of pathogens from people to the animals, a possible improvement could be to implement the washing stations and display signs explaining the visitors’ rules in the enclosure and to remind visitors to wash their hands before feeding the animals as shown by the management and enclosure analysis.

In the risk assessment step, we will focus on risk management actions, using a risk ranking score in line with the idea presented in Table 8, highlighting the importance of a stronger knowledge about the factors influencing the risk and the consequent possibilities to reduce them. We need to consider that the strength of knowledge is extremely important in the expression of judgments about factors and their related uncertainty, so the personnel conducting the assessment can affect the measures that should be implemented to mitigate the risk. Many measures could be taken for reducing the deviations in the assumption about the risk factors, but the obvious one is to look for personnel having specific competence on risk assessment and experience related to the animals. 

Also in the case of risk assessment, the procedure has to be tailored to the specific interaction and situation under scrutiny. For example, in AVIs in which only some of the animals of a stable social group participate and they are separated from the others in order to participate, a fourth scenario is likely to have to be included. This scenario recognises the same consequences as scenario 1 of the present pilot study, but has “being separated from other members of the same social group”, not “wrong handling/improper approach”, as exposure factors. Moreover, this further scenario involves both the animals who participate and those who are left behind. Moreover, in the present pilot study, the assessment compared consequences on welfare for a management model that provided for a direct human–animal contact to a management model which is the same in all relevant aspects apart from the fact that contact is absent, i.e., focused on the interactions itself. However, if some individual animals are kept in zoos only because of their involvement in AVIs (e.g., because they belong to a species which is not particularly attractive in an exhibit, but deemed easy to be handled by the public), the assessment should also take into account the general management of the animals under human care. In the latter case, the animals would not be under human management if not for the AVI, so the AVI is indirectly responsible for any risk to their welfare which comes from their being in the zoo (i.e., also for a risk due to unsuitable resting places, even if the resting places are not where the AVI takes place).

Faecal hormonal analyses [44] were not planned at the same time as the behavioural observations in this pilot study due to practical reasons, but a step assessing physiologic parameters is included in the theoretical template of the overall welfare assessment approach, acknowledging the importance to include physiologic parameters alongside behavioural ones when assessing animal welfare [41]. The results of the physiologic parameters assessment, for example an increase in cortisol levels, indicating an increase in arousal, would be then entered into the risk assessment procedure as further animal-based indicators of possible welfare risks. When assessing hormonal levels some problems could arise. For example, if faecal cortisol metabolites assessment were to be implemented in the present pilot study, some feasibility constraint would apply. At present, studies should still be carried out to identify safe and efficient faecal markers for giraffes, i.e., dietary additives that appear in faeces following ingestion and allow the researcher to assign faeces to the right animal when animals are kept in groups. Fuller et al. [45] were not able to identify a successful faecal marker for the giraffe because both blueberries (2 cups) or cooked corn or rice (2 cups) did not show in giraffes’ faeces. Although genetic analysis could identify to which individual the faecal sample belongs to [46], such methods tend to be expensive and are not devoid of technical problems, such as those highlighted by Creel and colleagues [47]. Another problem is that collecting the samples within a short time after defecation, in order to avoid changes in cortisol metabolites due to bacterial enzymes, as advised by Möstl and Palme [44], is not always possible, depending on housing and management of the individuals. So implemented, the welfare assessment approach hereby presented, is to be considered, as already mentioned, as the basis of a six-step dedicated protocol designed to assess the overall values of AVIs. In the protocol, the welfare assessment is a key aspect of the evaluation process against which all other possible outcomes of the interaction program, in terms of education and conservation, are evaluated. As the facilities offering AVIs vary in respect to their attention to animal welfare, conservation, education, as well as many other management issues, it is important to increase awareness on the need to implement a transparent, scientifically-based evaluation of AVIs whenever they are offered to the public. Therefore, there is a need to have an internationally recognised and standardised method that can be used to assess not only the possible effects of the interaction on the animals but also the overall value of such interactions. Thus, the multifaceted approach to animal welfare assessment hereby presented, once implemented on a practical level including physiologic parameters, shows promise to be useful for zoos, rescue centres, sanctuaries, and other facilities to assess their interaction programs and could be a model to assess also other management procedures.

## 5. Conclusions

As people have become more aware of issues related to animal welfare in zoos and other facilities, there is an increased need to evaluate AVIs on an integrated level, using a transparent analysis of all the aspects involved. In the present study, a multifaceted approach to animal welfare assessment during animal-visitor interactions, combining quantitative behavioural observations/analysis and a welfare risk-assessment procedure was presented, using the case of visitor-giraffe interaction as a pilot project. 

The presented multifaceted approach constitutes the basis of a six-step protocol designed to assess the value of interaction programs considering the impact both on animals and on visitors. The results presented in this study supported that the “Giraffe feeding” program could not be considered likely to negatively affect the welfare of the participating giraffes at all. On the contrary, the fact that they choose to participate suggests that they could perceive the interaction as positive. In particular, the giraffes did not show any behavioural change suggestive of an increased risk for their welfare due to the interaction program. The welfare risk analysis procedure identified an overall low risk for the welfare of the giraffes, mainly due to the absence of significant behavioural changes and to the fact that the enclosure analysis confirmed that the giraffes could choose whether to interact or not. As part of a protocol for the overall evaluation of animal-visitor interactions, the presented multifaceted approach, once implemented with an analysis of welfare relevant physiological parameters, shows promise to be useful for zoos, and other facilities allowing such interactions in order to assess their interaction programs and possibly other relevant management issues.

## Figures and Tables

**Figure 1 animals-08-00153-f001:**
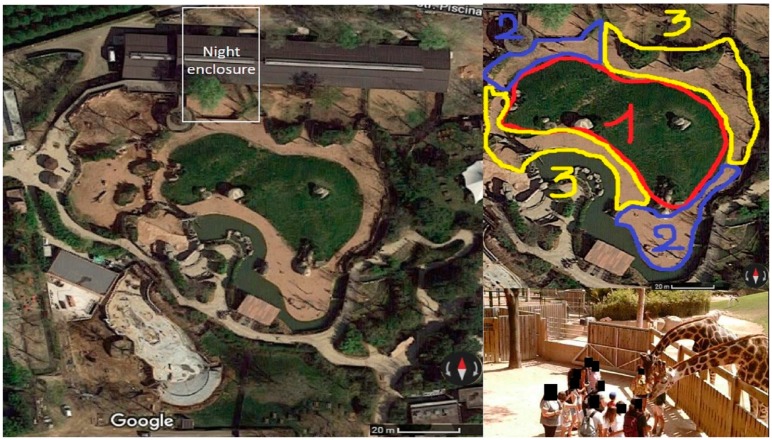
The giraffes’ enclosure: on the left—aerial view with the night enclosure in the grey rectangle; on the right, top—the zones used for space use analysis; on the right, bottom: a picture of an actual interaction.

**Table 1 animals-08-00153-t001:** Giraffes involved in the study.

Name	(Sub)Species	Age (Years) at the Beginning of the Study	Involvement in the Program
Baridi	*Giraffa camelopardalis antiquorum*	15	No
Karega	*Giraffa camelopardalis antiquorum*	8	No
Sam	*Giraffa camelopardalis rothschild*	13	Yes
Frantisek	*Giraffa camelopardalis rothschild*	7	Yes

**Table 2 animals-08-00153-t002:** Timing of the videotaped behavioural observations.

Session	Recording Schedule
“pre-” observation session	5 min for the first giraffe
	5 min for the second giraffe
	5 min for the third giraffe
	5 min for the fourth giraffe
pause	20–30 min.
“during” observation session	2 min for the first giraffe
	2 min for the second giraffe
	2 min for the third giraffe
	2 min for the fourth giraffe
	2 min for the first giraffe
	2 min for the second giraffe
	2 min for the third giraffe
	2 min for the fourth giraffe
	Continue to alternate every 2 min until the end of the interaction
pause	20–30 min.
“post-” observation session	5 min for the first giraffe
	5 min for the second giraffe
	5 min for the third giraffe
	5 min for the fourth giraffe

**Table 3 animals-08-00153-t003:** Working ethogram used in the study (adapted from Seeber and colleagues [13]).

Category	Behavior	Behavior Type	Description
Feeding	Browsing	State event	The giraffe uses tongue and/or lips to gather and ingest browse from trees or shrubs.
	Feeding	State event	The giraffe ingests food other than browse, such as concentrates or hay; captive animals are often fed on hay and concentrates from cribs or elevated feed buckets. Ingesting food offered by visitors during interaction is excluded as it constitutes the behaviour “being fed by visitors”.
	Drinking	State event	In order to drink from a water source on or below ground level, the giraffe has to splay its forelegs out laterally and flex its carpal joints to reach the water surface. Water is swallowed in this position as well.
	Being fed by visitors	State event	The animal eats from the hands of the interaction activity participants. The behaviour begins when the neck is protruding beyond the fence and lengthens the tongue to take the vegetables offered. It ends from the moment the giraffe raises his head.
	Other ingestion	State event	Includes other feeding behavior not previously listed: Graze, Geophagy, Osteophagy.
	Licking	State event	The giraffe licks the surface of an object. It can also be followed by nibbling on it. The tongue is presumably used to investigate the texture of an object.
Grooming	Self-grooming	State event	The animal cleans itself, licks or bites its body. It also includes ‘rub (object)’ [19], or the rubbing of the neck/head/body against an object.
Inactivity	Resting-Standing	State event	The animal is stationary, upright position. During this behaviour, the animal can ruminate ‘Ruminate’, ‘Scan’, or ‘Drowse’, given the difficulty on the identification of each of them singularly.
	Resting-Lying	State event	The body touches the ground with the giraffe lying in sternal position, legs are tucked in or folded up under the body. The head is carried with an erect or slightly bent neck. When lying down, front legs are bent first, followed the hind legs. In adults, it is only done for rather short periods of time, presumably to sleep, with the head resting on the body.
	Sleeping	State event	Lying on the ground in a sternal position (as when resting), neck curled, head resting on the animal‘s hip or thigh and eyes closed.
Locomotion	Walking	State event	The animal moves with a four-beat locomotion or with a three-beat movement with a moment of suspension (‘Canter’).
Social behaviours	Sparring	State event	This behaviour seems to be the counterpart of a fight, but sparring is considerably slower and less vigorous. Sparring develops slowly, sometimes initiated by necking and can continue over hours, and is usually interrupted oftentimes for several minutes to scan, or even to ruminate. The behaviour is performed by two, or up to eight individuals, who can stand parallel or antiparallel or in a different angle from each other.
	Licking urine	State event	The giraffe licks another giraffe‘s urine from the ground. Here we include also ‘urine testing’, which is performed by an adult after stimulating another to urinate. Licking urine in bulls is sometimes followed by a flehmen response.
	Mounting	State event	One animal stands right behind or on the side of another one, lifting its front legs on to conspecific‘s body, attempting to mount it. In adult bulls, the mounting attempt is usually preceded by pushing the other animal with the chest and lower neck. The mounted animal does sometimes not tolerate, but in other cases even ignores being mounted and continues feeding.
	Necking/Nuzzling/Rubbing	State event	Social behaviour that involves ‘Rubbing’ (against another animal) and ‘Necking’, when a giraffe rubs its head or neck against a conspecific’s body, sometimes leading to an entwining of the necks, and ‘Nuzzling’, a tactile encounter with conspecific by animal‘s nose or muzzle to conspecific’s nose or any other area then flanks or ano-genital area.
	Allo-Grooming	State event	One animal grooms another one’s body or crest by licking or biting.
Others	Other behaviours	State event	The animal is visible but is engaged in other behaviours than those listed above. This included other forms of stereotypies or other abnormal behaviour, agonistic behaviour, avoidance of contact/proximity with interacting visitors, escape attempts, getting away from interactions.
Not Visible	Not Visible	State event	The animal is not visible to the observer.

**Table 4 animals-08-00153-t004:** Checklist for management and enclosure analysis.

Management Checklist: Staff Actions and Procedures		
**Keepers and Staff Supervising Animal-Visitor Interactions**	**YES**	**NO**
Did the keeper inform visitors about the rules to be followed during the animal-visitor interaction? (e.g., how to touch the animals, not to smoke, drink and eat, act slowly, and do not yell, etc.)		
Did the keeper monitor animal behavior during the animal-visitor interaction?		
Did the keeper monitor visitors’ behavior during the animal-visitor interaction?		
Did the keeper recommend the visitor to wash their hands before animal-visitor interaction?		
Did the keeper quit any work activity in case of a serious uncontrolled risk is identified? (Suspect of disease or signs of irritability or aggression).		
Did the keeper have continuous training about biosecurity practices, zoonotic risk and appropriate practices to minimize these risks?		
Did the keeper have the knowledge of how to report exposures, accidents, injuries and illnesses?		
Did the keeper have continuous training about the procedures to avoid animal escape?		
Did the keeper have continuous training to recognize signs of health problems and stress in the animals held in the zoos?		
Did the keeper maintain suitable standards of hygiene to minimize the risk of disease transmission?		
Did the keeper assess and document the potential impact of each interactive experience on animal welfare and review it periodically?		
Did the keeper check that the animals are free of lesions/illness and/or disease before and after each animal-visitor interaction?		
**Veterinarians**	**YES**	**NO**
Are the veterinarians involved in the management decisions about the species and the individuals that participate in the interactions?		
Did the veterinarians compile and follow a preventive, curative, and nutritional veterinary program?		
Did the veterinarians perform zoonotic risk analyses?		
Did the veterinarians make sure that the food administered during the interaction is part of a nutritional veterinary program?		
Did the veterinarians update the clinical and pathological records?		
**Enclosure Checklist: Design, Construction and Procedure**	**YES**	**NO**
Was an animal interaction area, such as an animal enclosure where visitors can touch the animals, clearly defined?		
Were additional barriers present where the visitors pass to go into the enclosure (to avoid the escape of animals when the visitors enter)?		
Was a protocol to avoid escape defined?		
Were physical safety barriers present between the visitors and the animals during the interaction?		
Were unauthorised access prevented?		
Was the area of the enclosure where the interactions occurred well-ventilated?		
Was the area of the enclosure where the interactions occurred at least partially shaded?		
Did the area of the enclosure where the interaction occurred allow the animals to avoid the interaction, if they wished, without being followed by the public?		
Was the housing of the animals during interaction allow all of them to participate in the interaction if they wished?		
Were adequate signs present displaying visitors’ rules during interaction (not to smoke, eat, drink, proper hand washing, etc.)?		
Was the enclosure designed to allow correct cleaning and disinfection?		
Were hand-washing facilities available before accessing the interaction area?		
Was antibacterial hand gel available before accessing the interaction area?		
Were there limits to the number of participants per activity?		
Had an appropriate keepers/visitors/animals ratio been defined?		
Were there protocols for managing the biosecurity risks associated with visitor-animal interaction and emergency procedures?		

The Yes/No answers were given applying the following criteria: YES was chosen when the researcher had seen/heard the staff member performing the action (e.g., informing the public on what not to do during the interaction) or if the results of the staff member’s actions were evident (e.g., there were adequate signs displayed advising visitors not to smoke); NO when the consequences of the action the staff member should have performed were absent, suggesting no action was taken (e.g., there were no antibacterial hand gels available).

**Table 5 animals-08-00153-t005:** Animal welfare indicators for scenario 1 (adapted from a past paper [36]).

Title	Behavioural Observations
Scope	Animal-based measure: *Giraffa Camelopardalis*
Sample size	Four animals
Method description	According to ethogram-(other behaviours: escape behavior/avoidance behaviour).
Classification	Individual level:
	1—No evidence of behaviours reflecting the worsening of subjective experiences;
	2—Evidence of behaviours reflecting the worsening of subjective experiences.

**Table 6 animals-08-00153-t006:** Animal welfare indicators for scenario 2 (adapted from a past paper [36]).

Title	Skin Lesions
Scope	Animal-based measure: *Giraffa camelopardalis*
Sample size	Four animals
Method description	These injuries can be caused by wrong handling or improper approach by the visitor.
	Conduct an external visual examination before and after the interaction session.
	The skin of the animals must not show injuries or abnormalities.
	Observe the anatomical regions of the head and neck.
Classification	For each animal, the extent and severity of the lesion must be defined by assigning the following scores:
	1—No obvious injuries.
	2—Injury involving a limited area of the surface, without compromising the deep layers.
	3—Large lesion, involving deeper layers and possibly aggravated by an inflammatory state.

**Table 7 animals-08-00153-t007:** Animal welfare indicators for scenario 3 (adapted from a past paper [36]).

Title	Zooanthroponosis
Scope	Animal-based measure: *Giraffa camelopardalis*
Sample size	Four animals
Method description	In-depth diagnostic tests following detection of symptoms attributable to infective or diffusive diseases to determine the causes and ascertain their zoonotic origin.
Classification	1—No diagnosis of infectious disease;
	2—Diagnosis of a zoonotic infectious disease presenting with impairment of general health, asthenia, anorexia, or involvement of a single organ, without severe complications;
	3—Situation of a serious infection that involved multiple organs or with severe complications.

**Table 8 animals-08-00153-t008:** Welfare risk assessment score.

Risk Ranking	Welfare Score	Description and Outcomes—Welfare Score Response
Negligible	0	No issue—indirect monitoring only—e.g., via annual reports (and photos/video footage, if deemed necessary).
Low	1–4	Acceptable—indirect monitoring only—e.g., via annual reports (and photos/video footage, if deemed necessary).
Moderate	5–11	Active management required (improvement of objectives that can be planned over time without urgency).
High	>12	Urgent action required.

**Table 9 animals-08-00153-t009:** Values of welfare score (FE stands for Frequency of exposure, FC stands for Frequency of consequences; WS stands for Welfare score).

Exposure Assessment		Consequences Characterization				Risk Characterization
Hazard Description	FE	Animal-Based Indicators	Severity	Duration	FC	WS
Wrong handling/improper approach (scenario 1)	1	Behavioural observation	1	1	2	2
Wrong handling/improper approach (scenario 2)	1	Skin lesion	1	1	2	2
Effective contact with a zoonotic agent (scenario 3)	2	Zooanthroponosis	1	1	2	4

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
