# Peer review of "Assessing Animal Welfare in Animal-Visitor Interactions in Zoos and Other Facilities. A Pilot Study Involving Giraffes"

_animals, 2018, doi:10.3390/ani8090153_

Round 1
Reviewer 1 Report
Since physiological measures were not possible in this study, it is difficult to know if subjects were disturbed by the experience of human contact. However, the point is well made that the animals were not forced to participate and expressed a calm demeanor. The study is important in that so many zoos are utilizing giraffe in this way suggesting a need to evaluate the experience. This pilot study should be replicated with physiological measures.
Author Response
Since physiological measures were not possible in this study, it is difficult to know if subjects were disturbed by the experience of human contact. However, the point is well made that the animals were not forced to participate and expressed a calm demeanor. The study is important in that so many zoos are utilizing giraffe in this way suggesting a need to evaluate the experience. This pilot study should be replicated with physiological measures.
Thank you for your comments, we plan to replicate it including faecal cortisol metabolites assessment and with more than 7 episodes per condition as soon as we have funds. The (more scientific) problems we are trying to tackle concerning faecal sampling are briefly discussed in the discussion (lines 578-588).
Reviewer 2 Report
The study focuses in an interesting research topic: the effect of visitor-giraffe interaction on animal welfare. However, the article has serious flaws and research has not been conducted correctly.
No results or conclusions should be presented in the introduction section.
The research study is presented as a combination of three aspects: quantitative behavioural observations, endocrine parameter evaluation and welfare risk assessment procedure. However, none endocrine parameter has been assessed. Even if it is proposed for future studies it should not be included in the abstract, introduction, material and methods, and conclusion; nor be presented as part of the current study.
Several behaviours are observed (according to the ethogram) but only the ones related with feeding behaviour are discussed or shown in the results section. No results are shown for the use of space and ambient temperature.
The welfare risk assessment is confusing and difficult to understand.
Some results and conclusions are not correctly reached or confusing. Some examples follow.
Four animals are presented in this study, but only two interact with the visitors. Observations are done in three episodes (pre-, during, and post-interaction giraffe-visitor) but only results of one episode are presented (during).
Line 286: ‘As logical, the giraffes ate more during interaction than control episodes’. Line 292: ‘The giraffes showed total ingestion behaviour longer during interaction vs control episodes’. The way ‘eating’ is defined on the ethogram makes this behaviour impossible to happen in control episodes (where no interaction with visitor happens).
Lines 345-347: ‘Unsurprisingly, behavioural observations/analysis did not identify changes in the giraffes’ behaviour suggestive of an altered welfare level. It has long been known that, if animals have the possibility to control events, they tend to not perceive them, even negative ones, as stressors’. Line 427: ‘the fact that they choose to participate suggests that they could perceive the interaction as positive’. I do not think that with the methodology and the results presented in this article there is enough information to reach these conclusions and make these statements.
These are the main reasons why I do not consider this article suitable for publication in Animals.
Author Response
The study focuses in an interesting research topic: the effect of visitor-giraffe interaction on animal welfare. However, the article has serious flaws and research has not been conducted correctly.
No results or conclusions should be presented in the introduction section.
The guidelines of the Journal (the part concerning introduction is pasted below) explicitly ask for aim and conclusions to be mentioned at the end of the introduction
· “Introduction: The introduction should briefly place the study in a broad context and highlight why it is important. It should define the purpose of the work and its significance, including specific hypotheses being tested. The current state of the research field should be reviewed carefully and key publications cited. Please highlight controversial and diverging hypotheses when necessary. Finally, briefly mention the main aim of the work and highlight the main conclusions. Keep the introduction comprehensible to scientists working outside the topic of the paper.”
The research study is presented as a combination of three aspects: quantitative behavioural observations, endocrine parameter evaluation and welfare risk assessment procedure. However, none endocrine parameter has been assessed. Even if it is proposed for future studies it should not be included in the abstract, introduction, material and methods, and conclusion; nor be presented as part of the current study.
Now the endocrine parameter assessment is mentioned only in the discussion (lines 574-588)
Several behaviours are observed (according to the ethogram) but only the ones related with feeding behaviour are discussed or shown in the results section.
We presented the only behavior which showed a statistical difference between control and interaction episodes. It varied only during the DURING session and only for the two giraffes participating. Now we specify that no other differences were found (other behaviors, other sessions, other giraffes) (lines 394-400)
No results are shown for the use of space
Now we detail the results for the use of space (lines 401-404)
and ambient temperature.
Ambient temperature was measured only to verify that it did not differ significantly between interaction and control episodes. Although this is a very interesting suggestion for a further study, assessing differences in behavior corresponding to different temperatures was beyond the scope of the paper and would have required a slightly different study design, at least one with more episodes (lines 191-193)
The welfare risk assessment is confusing and difficult to understand.
We rewrote part of it in order to try to make it clearer (sections 2.4*)
Some results and conclusions are not correctly reached or confusing. Some examples follow.
Four animals are presented in this study, but only two interact with the visitors.
Two animals never interact. We deemed it unethical and confounding to try to make them interact if they do not choose to do so. A discussion on the possible causes of this choice is now added in the discussion (lines 509-522).
Observations are done in three episodes (pre-, during, and post-interaction giraffe-visitor) but only results of one episode are presented (during).
A sentence saying that there were no difference in behavior in the pre and in the post session is now added (lines 394-400)
Line 286: ‘As logical, the giraffes ate more during interaction than control episodes’. Line 292: ‘The giraffes showed total ingestion behaviour longer during interaction vs control episodes’. The way ‘eating’ is defined on the ethogram makes this behaviour impossible to happen in control episodes (where no interaction with visitor happens).
We changed “eating” into “being fed by visitors” (e.g., line 384)
Lines 345-347: ‘Unsurprisingly, behavioural observations/analysis did not identify changes in the giraffes’ behaviour suggestive of an altered welfare level. It has long been known that, if animals have the possibility to control events, they tend to not perceive them, even negative ones, as stressors’. Line 427: ‘the fact that they choose to participate suggests that they could perceive the interaction as positive’. I do not think that with the methodology and the results presented in this article there is enough information to reach these conclusions and make these statements.
As the reason for which the methodology and the results presented in this article are not enough to reach the conclusion is not explained, we are unable to answer to this comment, or to correct the manuscript or even to design our future studies better, we apologise for that.
These are the main reasons why I do not consider this article suitable for publication in Animals.
Reviewer 3 Report
This paper presents a relatively straightforward process for at least partly evaluating the possible welfare impacts of animal-human interactions. I say partly because 1) the behavioral measures chosen will vary from interaction to interaction and may/may not be appropriate and 2) in this case only behavior work was done, and 3) actual changes in animal subjective experiences (positive or negative) are not defined. So this process gives information more about risk than actual welfare. This doesn't make it useful but it should be clearly expressed that way so readers don't misinterpret the intent of the tool. I appreciate the authors' interest in presenting a transparent tool; however, it is really no less subjective than other less formal, less "process heavy" methods. Changing subjective assessments of frequency/duration/intensity etc. for example into ordinal scale numbers really does not make any of this more objective. So I would like to see the authors better clarify that this is a subjective process, not really an objective one except for the behavior study portion, which will take a multitude of different forms for different interactions. The risk assessment portion of this is virtually subjective in my view. That doesn't make it less valuable, but it needs to be better acknowledged. My own personal bias though is that this a lot of "process" for matters than can be just as well handled with conversation. Some specific comments follow, but generally there are lots of spelling and grammar errors - too many to identify individually:
Line 48-49, not necessary to call out these kinds of experiences here
Table 1: the source and genetics of the animals really isn't relevant here
line 150 how was visitor number at the enclosure ACTUALLY gathered/calculated? This doesn't seem to show up elsewhere though so I'm not sure whether it is necessary to include. There seems to be lots of detail in the methods that doesn't really surface anywhere later in the paper.
157: flooding?
Many of the tables are too narrow in the pdf for review so there are lots of words cut off or split between lines.
180-181 delete text in parentheses
188-197. No need to have this in methods when you didn't actually do it. the way it is described in the discussion is good enough
Table 4: the keeper and vet sections are written as questions whereas the facility section is written as a "to do" list. wording should be consistent.
Figures 1 and 2 really aren't that helpful, interesting, or necessary.
239-241 - delete text in parenthesis
Tables 5-7 are not necessary and should just be conveyed in text. These are some aspects of the process that contribute to its significant subjectivity too. Also I'm not sure why frequency of exposure is given the symbol PF, same comment for PC. Doesn't make sense to me in english.
Tables 8-10 also seem at least partially redundant with parts of the text. I don't feel they are necessary.
Table 11. It isn't really clear how the numerical range from 0 to >15 was developed. There also seems to be no real operational difference between the "high" and "extreme" ratings. One could even argue that they don't differ from the "moderate" either...
293-294. space use should be quantified and presented that way
295-298 maybe here just say that these observations are from notes.
321 how exactly did you decide there were no welfare consequences. So you measured behavioral changes, whether or not injuries occurred or were likely and whether or not zoonoses were likely (but not really whether they occurred since the time frame was short)? You didn't actually measure welfare, you measured welfare indicators.. and these vary based on individual philosophy about welfare. Earlier in the paper, subjective experiences were mentioned but you actually didn't measure any indicators of subjective experiences, except possibly some behaviors. So again I think your process helps you identify welfare risk but not actual welfare level.
Table 12 I think you need to label the columns better so we know which ones correspond to PF, MS, and PC since that is your formula.. I also don't know why you refer to this formula as linear programming.. it is just a simple multiplication.
379-382. There is a lot of jargon here.. need to clarify language.
Author Response
This paper presents a relatively straightforward process for at least partly evaluating the possible welfare impacts of animal-human interactions. I say partly because 1) the behavioral measures chosen will vary from interaction to interaction and may/may not be appropriate
We discussed the need to custom the methods to the interaction one studies in section 2.3, (very briefly) at the beginning of the discussion (lines 461-462) and on lines 500-502.
and 2) in this case only behavior work was done, and 3) actual changes in animal subjective experiences (positive or negative) are not defined. So this process gives information more about risk than actual welfare. This doesn't make it useful but it should be clearly expressed that way so readers don't misinterpret the intent of the tool.
This is now stated in section 2.3 and in the discussion (lines 488-489), the sentence has been rephrased into “changes in the giraffes’ behaviour suggestive of altered risk to the welfare level of the animals”
I appreciate the authors' interest in presenting a transparent tool; however, it is really no less subjective than other less formal, less "process heavy" methods. Changing subjective assessments of frequency/duration/intensity etc. for example into ordinal scale numbers really does not make any of this more objective. So I would like to see the authors better clarify that this is a subjective process, not really an objective one
And
The risk assessment portion of this is virtually subjective in my view. That doesn't make it less valuable, but it needs to be better acknowledged. My own personal bias though is that this a lot of "process" for matters than can be just as well handled with conversation.
It has been acknowledged in lines 276-278
except for the behavior study portion, which will take a multitude of different forms for different interactions.
We discussed the need to custom the methods to the interaction one studies in section 2.3, (very briefly) at the beginning of the discussion (lines 461-462) and on lines 500-502.
Some specific comments follow, but generally there are lots of spelling and grammar errors - too many to identify individually:
We read it again looking for and correcting the mistakes we found
Line 48-49, not necessary to call out these kinds of experiences here
Deleted
Table 1: the source and genetics of the animals really isn't relevant here
The columns were eliminated
line 150 how was visitor number at the enclosure ACTUALLY gathered/calculated? This doesn't seem to show up elsewhere though so I'm not sure whether it is necessary to include. There seems to be lots of detail in the methods that doesn't really surface anywhere later in the paper.
We have simplified the sentence, citing only what is important for methods and/or is reported in the results (lines 185 to 188) and explaining the reasons for recording those variables (lines 185 to 193)
157: flooding?
An explanation of flooding (i.e., forceful exposure of the animal to the possibly fear inducing stimuli at a high intensity) is now on lines 201-202
Many of the tables are too narrow in the pdf for review so there are lots of words cut off or split between lines.
I had a co-author also correct all formatting. I do not know why but this file gave us a lot of problems of formatting.
180-181 delete text in parentheses
deleted
188-197. No need to have this in methods when you didn't actually do it. the way it is described in the discussion is good enough
Now the endocrine parameter assessment is mentioned only in the discussion (lines 574-588)
Table 4: the keeper and vet sections are written as questions whereas the facility section is written as a "to do" list. wording should be consistent.
Amended
Figures 1 and 2 really aren't that helpful, interesting, or necessary.
They have been eliminated
239-241 - delete text in parenthesis
deleted
Tables 5-7 are not necessary and should just be conveyed in text. These are some aspects of the process that contribute to its significant subjectivity too. Also I'm not sure why frequency of exposure is given the symbol PF, same comment for PC. Doesn't make sense to me in English.
Tables 5-7 were deleted and covered in text, and acronyms were changed (lines 335-348)
Tables 8-10 also seem at least partially redundant with parts of the text. I don't feel they are necessary.
As other referees commented on the risk assessment not being clear enough, we tried to better contextualize the tables in the context of welfare quality, but we did not eliminate them, because they appear as an clearer way to visualize things
Table 11. It isn't really clear how the numerical range from 0 to >15 was developed. There also seems to be no real operational difference between the "high" and "extreme" ratings. One could even argue that they don't differ from the "moderate" either...
The table (now table 8) has been modified according to this comment
293-294. space use should be quantified and presented that way
Space use is now presented in the results lines 401-403.
295-298 maybe here just say that these observations are from notes.
The adding up was done on an excel file (now specified on line 388), notes were used for other abnormal behavior, agonistic behavior and avoiding/escaping (lines 211-218 )
321 how exactly did you decide there were no welfare consequences. So you measured behavioral changes, whether or not injuries occurred or were likely and whether or not zoonoses were likely (but not really whether they occurred since the time frame was short)? You didn't actually measure welfare, you measured welfare indicators.. and these vary based on individual philosophy about welfare. Earlier in the paper, subjective experiences were mentioned but you actually didn't measure any indicators of subjective experiences, except possibly some behaviors. So again I think your process helps you identify welfare risk but not actual welfare level.
This is now stated in section 2.3 and in the discussion (lines 488-489), the sentence has been rephrased into “changes in the giraffes’ behaviour suggestive of altered risk to the welfare level of the animals”
Table 12 I think you need to label the columns better so we know which ones correspond to PF, MS, and PC since that is your formula.. I also don't know why you refer to this formula as linear programming.. it is just a simple multiplication.
The column labels have been changed, and the sentence rephrased eliminating “linear programming”
379-382. There is a lot of jargon here.. need to clarify language.
The sentences have been rephrased to be clearer (lines 531-539)
Reviewer 4 Report
This manuscript presents a partial overview of a protocol to assess
the impacts of visitor-animal interactions in zoos. The overall
concept, with all six factors included, is intriguing, but it is unclear
how all six components will be used in concert to provide assessments.
As currently presented in the manuscript, the behavior evaluation and
management/enclosure assessment are not very novel. The way the risk analysis may be a novel way of putting information together, but is not described in a way that makes it compelling to do so.The authors describe
endocrine parameters as a third measure, but then reveal this was not
included in the study. It would be more appropriate to remove it from
the manuscript except where it is mentioned in the discussion.
There are some grammatical and formatting errors, as well as missing words, that should be corrected (e.g., lines 80, 89, 93, 211, labels on figure 2). There
is some information missing from the methods which would be helpful to
the reader. Why did only two of the giraffes participate in the feeds?
Was this by choice? How does the experience differ for the giraffes
between interactions with different numbers of visitors. Do fewer
participants mean shorter duration and less food? Was this controlled
for? The behavioral observations during feeds include rotation between
giraffes every two minutes. What if the encounter lasts less than 8
minutes due to low number of participants? No abnormal or agonistic
behaviors are listed in the ethogram. These seem particularly relevant
and should be included. Did the researchers record if the giraffes ever
broke away or refused to participate? A schematic of the habitat would
be helpful.
It is unclear how identifying the exposure scenarios and the welfare consequences and measurements differs as explained in the text (lines 230-236). The same information is repeated in both sections and no measurements are described. It would be more helpful to actually describe a process that can be applied to other situations.
To
calculate the welfare indicators for table 8, it is not clear what
behaviors were utilized. Was it just escape/avoidance behaviors? These
are not included in the ethogram, but should be. There are additional
behavioral measures that could reflect a change in internal state, but
this table does not indicate what has been considered.Table 8 also only
provides tow choices to quantify severity of consequences for the first
scenario. This is a very vague scale that could use refinement to be
more beneficial.
No data is shown for use of space. More results should be included on this topic.
Consistency
in terminology is important. The term severity and intensity seem to be
used interchangeably (lines 255 and 276). It is also not entirely well
explained why the researchers chose to calculate the welfare score the
way they did. It is unclear how this approach is different from
using science to assess a situation (lines 381-382). Data need to be collected and used to come to a conclusion and take
action.
More recent work has been done on human-animal
interactions than what is referenced in the manuscript. One study
specifically addresses the impact of feedings on giraffes (Orban,
Siegford and Snyder, 2016), and those results could be helpful to the giraffe study in this manuscript.
Overall, what is
reported here does not seem vastly different from how welfare is being
assessed by many and the protocol described does not seem any less
cumbersome or more innovative. Some steps even seem extraneous, like
building the conceptual models. It would be more beneficial to describe
the process in such a way that it helps one identify factors and
consequences and how to apply the process to their own situations. As of
now, the manuscript seems a bit torn between trying to describe a
process that could theoretically be applied to a variety of situations
(if described in such a way) and providing results of a project on the impact
of visitor feeding on giraffes. The giraffe study comes across as less
of an example of how to apply the process and more a central focus of
the paper. Finding systematic ways to assess welfare, including the impact the visitor-animal interactions, is critical and the authors' aim to do so is commendable. Rethinking how this manuscript convinces readers of the value of their approach is recommended.
Author Response
This manuscript presents a partial overview of a protocol to assess the impacts of visitor-animal interactions in zoos. The overall concept, with all six factors included, is intriguing, but it is unclear how all six components will be used in concert to provide assessments.
More detail on the six steps are given in the introduction lines 74-82
As currently presented in the manuscript, the behavior evaluation and management/enclosure assessment are not very novel. The way the risk analysis may be a novel way of putting information together, but is not described in a way that makes it compelling to do so.
We rewrote most of the parts about risk assessment in an attempt to make it clearer (sections 2.4*)
The authors describe endocrine parameters as a third measure, but then reveal this was not included in the study. It would be more appropriate to remove it from the manuscript except where it is mentioned in the discussion.
Now the endocrine parameter assessment is mentioned only in the discussion (lines 574-588)
There are some grammatical and formatting errors, as well as missing words, that should be corrected (e.g., lines 80, 89, 93, 211, labels on figure
We corrected all the mistakes we found
2). There is some information missing from the methods which would be helpful to the reader. Why did only two of the giraffes participate in the feeds? Was this by choice?
A discussion on the possible causes of this is now added in the discussion (lines 509-522).
How does the experience differ for the giraffes between interactions with different numbers of visitors. Do fewer participants mean shorter duration and less food? Was this controlled for? The behavioral observations during feeds include rotation between giraffes every two minutes. What if the encounter lasts less than 8 minutes due to low number of participants?
Some more details on the number of participants and the duration of the interactions are given in the results. The interaction always lasted at least 8 minutes and there was no correlation between their duration and number of participants (lines 404-407).
Although the suggestion to investigate changes in behavior linked to the different number of visitors is a very interesting one, an analysis of the effects number of visitors on the behaviour of the giraffes was outside the scope of the present study, also because it would have needed a much bigger sample size (lines 191-193)
No abnormal or agonistic behaviors are listed in the ethogram. These seem particularly relevant and should be included. Did the researchers record if the giraffes ever broke away or refused to participate?
All these behaviors, apart from licking the tree bark (which could be stereotypic), were never seen (as now said in lines 397-400). We detailed how we methodologically handled those behaviors in lines 211-218
A schematic of the habitat would be helpful.
Fig 1 has been added
It is unclear how identifying the exposure scenarios and the welfare consequences and measurements differs as explained in the text (lines 230-236). The same information is repeated in both sections and no measurements are described. It would be more helpful to actually describe a process that can be applied to other situations.
The sentence “An exposure scenario is a sequence or combination of events in relation to the risk question that includes in general, all information on the events to which animals of the target population are subjected. In this phase, relevant combinations of the identified factors and their exposure levels are defined.” has been added to clarify the issue (lines 306-308)
To calculate the welfare indicators for table 8, it is not clear what behaviors were utilized. Was it just escape/avoidance behaviors? These are not included in the ethogram, but should be.
Identification of the behaviors has been included in the table (5). We detailed how we methodologically handled escape/avoidance behaviors in lines 211-218
There are additional behavioral measures that could reflect a change in internal state, but this table does not indicate what has been considered.
Identification of the used behaviors has been included.
Table 8 also only provides tow choices to quantify severity of consequences for the first scenario. This is a very vague scale that could use refinement to be more beneficial.
We agree with you, and it is our intention to refine the scale in the future, also including the possibility to evaluate the possible enriching value of, but because it is the first pilot of implementation, wanted to keep it simple. Also, we felt that more knowledge on validated correlates of positive and negative mental states has to be available for this species in order to be able to modulate this scale better
No data is shown for use of space. More results should be included on this topic.
Space use is now presented in the results lines 401-403
Consistency in terminology is important. The term severity and intensity seem to be used interchangeably (lines 255 and 276).
amended
It is also not entirely well explained why the researchers chose to calculate the welfare score the way they did. It is unclear how this approach is different from using science to assess a situation (lines 381-382). Data need to be collected and used to come to a conclusion and take action.
The sentence “The values of the welfare score and the related actions to be taken derive from combinatorial simulations between the exposure elements and the consequences in order to provide management measures compatible and consistent with built scenarios, in accordance with the “as low as reasonably practicable” - ALARP principle [38]” was added to clarify the issue (lines 375-379).
More recent work has been done on human-animal interactions than what is referenced in the manuscript. One study specifically addresses the impact of feedings on giraffes (Orban, Siegford and Snyder, 2016), and those results could be helpful to the giraffe study in this manuscript.
A discussion of the results as compared with Orban, Siegford and Snyder, 2016 has been added in the discussion (lines 500-508)
Overall, what is reported here does not seem vastly different from how welfare is being assessed by many and the protocol described does not seem any less cumbersome or more innovative. Some steps even seem extraneous, like building the conceptual models. It would be more beneficial to describe the process in such a way that it helps one identify factors and consequences and how to apply the process to their own situations. As of now, the manuscript seems a bit torn between trying to describe a process that could theoretically be applied to a variety of situations (if described in such a way) and providing results of a project on the impact of visitor feeding on giraffes. The giraffe study comes across as less of an example of how to apply the process and more a central focus of the paper. Finding systematic ways to assess welfare, including the impact the visitor-animal interactions, is critical and the authors' aim to do so is commendable. Rethinking how this manuscript convinces readers of the value of their approach is recommended.
We modified M&M (e.g., adding section 2.3 and section 2.4) and rewrote the discussion in order to give priority to the process, and to discuss the results of the study as an example of the process not as a study per se (e.g., lines 468-473; 523-527; 577-588)
Round 2
Reviewer 2 Report
The paper has been considerably improved! It is better aimed and interesting to read.
I only have a few suggestions, more related with form than content.
The hours of events and observations are most of the times expressed with am/pm, but some hours do not have am/pm and I think it should be added to be more consistent.
About citation, since in this journal the references have to be cited like [n], I think that some edition should be done in certain parts of the text. For example, the sentences in lines 229, 244, 251, 253, 397, 399 and 401 (and maybe some others that passed undetected) are finished abruptly with the [n]. It would sound better if instead of saying '...adapted from that in [19]' or '...the same approach as in [21]', something like the following was written: '...adapted from that in Seeber and colleagues [19]' or '...the same approach as in Bertocchi and colleagues [21]'.
I detected lack of spaces between some words (might be more in the text):
Line 259 'above mentioned', 549 'has long' and 559 'above mentioned'
In lines 323, 414, 469, 513, 603 (maybe there are more) instead of Tab. should appear Table. Also, in line 121 figure should be spelled with capital F (Figure).
Author Response
The paper has been considerably improved! It is better aimed and interesting to read.
I only have a few suggestions, more related with form than content.
The hours of events and observations are most of the times expressed with am/pm, but some hours do not have am/pm and I think it should be added to be more consistent.
Hour format has been standardized
About citation, since in this journal the references have to be cited like [n], I think that some edition should be done in certain parts of the text. For example, the sentences in lines 229, 244, 251, 253, 397, 399 and 401 (and maybe some others that passed undetected) are finished abruptly with the [n]. It would sound better if instead of saying '...adapted from that in [19]' or '...the same approach as in [21]', something like the following was written: '...adapted from that in Seeber and colleagues [19]' or '...the same approach as in Bertocchi and colleagues [21]'.
The names of the authors were added on lines: 45, 182-183, 210-211, 231-233, 484, 614, 616.
I detected lack of spaces between some words (might be more in the text):
Line 259 'above mentioned', 549 'has long' and 559 'above mentioned'
We are sorry, now we have checked again for them, and amended them. Unluckily every time the file is opened using a different computer (we are several authors, located in different places, working on it using different computers), spaces disappear in multiple random points in the text (and sometimes also some words change font or size), so it has been nightmarish.
Reviewer 4 Report
The revision and reorganization of the manuscript has improved the overall readability and flow. There are corrections needed throughout for punctuation errors and
missing spaces between words, so another careful read-though is
recommended. The process for determining how to best apply the risk assessment to various situation is still somewhat vague and subjective (which is not the same as qualitative). More examples, aside from the pilot testing with the giraffes, would be helpful within the descriptions of the various steps. This is still a relatively cumbersome system, and more clear explanations in some areas could help with this and how readers could apply it more easily to their situations. However, organizing multiple types of information together certainly helps not only synthesize the information, but also assists with decision making from the management perspective.
Author Response
The revision and reorganization of the manuscript has improved the overall readability and flow. There are corrections needed throughout for punctuation errors and missing spaces between words, so another careful read-though is recommended.
We are sorry, now we have checked again for mistakes and missing spaces, and amended them. Unluckily every time the file is opened using a different computer (we are several authors, located in different places, working on it using different computers), spaces disappear in multiple random points in the text (and sometimes also some words change font or size), so it has been nightmarish.
The process for determining how to best apply the risk assessment to various situation is still somewhat vague and subjective (which is not the same as qualitative).
We absolutely agree that qualitative does not equate to subjective, although there is still some misinterpretation about this in the field
More examples, aside from the pilot testing with the giraffes, would be helpful within the descriptions of the various steps. This is still a relatively cumbersome system, and more clear explanations in some areas could help with this and how readers could apply it more easily to their situations. However, organizing multiple types of information together certainly helps not only synthesize the information, but also assists with decision making from the management perspective
We added some examples of parameters which could be used in step 1 (and of situations in which they are suitable) on lines 522-535, examples of differences in risk assessment (and of situations in which they are suitable) on lines 582-598, and an example for physiologic parameters on lines 603-605. We hope we managed to make the process clearer.